# “Back into Life—With a Power Wheelchair”: Learning from People with Severe Stroke through a Participatory Photovoice Study in a Metropolitan Area in Germany

**DOI:** 10.3390/ijerph191710465

**Published:** 2022-08-23

**Authors:** Tabea Böttger, Silke Dennhardt, Julia Knape, Ulrike Marotzki

**Affiliations:** 1Institute of Health Science, Faculty of Medicine, University of Lübeck, 23562 Lübeck, Germany; 2Faculty of Social Work and Health, University of Applied Sciences and Arts Hildesheim, Holzminden, Göttingen (HAWK), 31134 Hildesheim, Germany; 3Physio- and Occupational Therapy Program, Faculty of Health, Alice Salomon Hochschule Berlin (ASH), University of Applied Sciences, 12627 Berlin, Germany; 4Independent Researcher, 10439 Berlin, Germany

**Keywords:** participatory research, photovoice, power wheelchair, community mobility, social participation, stroke, rehabilitation

## Abstract

Severe stroke leads to permanent changes in everyday life. Many stroke survivors depend on support in community mobility (CM). This leads to restrictions and limited social participation. A power wheelchair (PWC) can enable independent CM and reduce such restrictions. This participatory study focused on how people with severe stroke experience their CM in a PWC in Berlin/Germany and what changes they want to initiate. A research team of five severe stroke survivors and two occupational therapists examined the question using photovoice. Stroke survivors took photos of their environment, presented, discussed, and analyzed them at group meetings to identify themes, and disseminated their findings at exhibitions and congresses. The photos emphasize the significance of and unique relationship to the PWC for the self-determined expression of personal freedom. As a complex, individualized construct, CM requires an accessible environment and diverse planning strategies by PWC users to arrive at their destination and overcome suddenly occurring obstacles. Desired changes stress CM independent of external help, increased social esteem, and active involvement in the provision of assistive devices. Voices of severe stroke survivors need to be heard more in healthcare and research to ensure the possibility of equal social participation.

## 1. Introduction

For people who suffer a brain injury, life changes suddenly and unexpectedly. Stroke, as a form of acquired brain injury, is a major cause of disability in adults worldwide [1,2], and due to demographic changes, the absolute number of people affected is continuously increasing [3]. Experiencing a stroke also affects one’s mobility; for about two-thirds of stroke patients, mobility is initially limited [4,5]. Three months after the acute event, about 20% of those affected are still dependent on a wheelchair, and in about 70%, gait speed and endurance are reduced to an extent relevant to everyday life [5]. Around 14–31% of stroke survivors are severely affected because they have multiple functional impairments and remain dependent on assistance in activities of daily living (ADLs) and mobility, regardless of rehabilitation efforts [6].

After a stroke or traumatic brain injury, many people are able to move independently again within their own home, but relating to community mobility (CM), they are dependent on support [7,8,9,10,11]. A recent review [7] about CM after stroke concludes that moderate to severe limitations in CM persist up to and beyond four years after stroke. It showed that all domains of the International Classification of Functioning, Disability, and Health (ICF) [12] affected the ability to reach CM, “including body structures and function, activity performance, as well as personal and environmental factors” [7] (p. 12). Even though CM encompasses all modes of functional mobility, including mobility in a (power) wheelchair [13], studies rarely take this into account [7]. Thus, the current review could not include any study in which stroke survivors used alternative means of transport (e.g., wheelchair) or had communicative and cognitive impairments, as this was an exclusion criterion in all studies (ibid.). This example shows that there is little evidence to date that addresses CM in stroke survivors who use other means of transportation, such as a PWC, and/or have severe impairments.

As a central connecting function in everyday life, CM takes on and facilitates social participation. It enables us to leave our homes to go grocery shopping or to work, meet friends, travel—and much more [8]. Experiencing limitations in social participation and choice of activities that were taken for granted before the stroke has a huge negative impact on health and life satisfaction [14,15]. The loss of activities and occupations, specifically outside the house, often leads to a sense of dependence [16,17], social isolation, and being bound to one’s home [9,18,19].

The use of a mobility assistive device, especially a wheelchair, can be an important “enabler of community participation” [20] (p. 18) for stroke survivors. A power wheelchair (PWC) or a scooter may have various positive effects, such as being able to transport oneself independently to places that enable participation in leisure and social activities and, thus, role enablement [21,22,23,24]. Studies point out the high satisfaction of users with their PWC, mainly due to the created independent mobility outside the home [22,23,25,26]. These positive results are also evident in small initial studies with people after acquired brain injury [20,21,24]. First studies show that limitations in activities and participation reported by stroke survivors can be reduced in most cases by the provision of a PWC [20,24]. However, stroke survivors experience many environmental barriers when using a PWC, such as the nature and layout of buildings as well as personal attitudes towards assistive devices (ibid.). Most of the studies involved people with an average age of 67 years or older and who had no or only few communication and cognitive impairments. Less is known about younger stroke survivors (≤55 years) with communication and cognitive impairments, how they use their PWC in CM, and how satisfied they are.

In Germany, the provision process of assistive devices is an integral part of the German healthcare system and neurological rehabilitation [27,28]. Nevertheless, there is little data about provision processes, frequency of use, and satisfaction with assistive devices available [28,29,30], and various problems in the provisioning process are discussed, such as by the German Society for Rehabilitation (Deutsche Vereinigung für Rehabilitation e.V., DVfR) [31,32]. The very few existing studies reveal that PWCs are seldom or not at all prescribed for neurological rehabilitation in Germany [28,33,34]. The possibilities for successful social participation, which can be especially facilitated by a PWC, are not yet sufficiently considered by decision-makers and health care professionals involved in the care process [32]. This shows the need for further research that focuses on this issue.

According to the International Classification of Functioning, Disability and Health (ICF) [12] and the UN Convention on the Rights of Persons with Disabilities (UN CRPD) [35], effective and equal participation of people with disabilities in social life is an aim of the rehabilitation process that has to be ensured [29,36]. This reorientation also requires a practice-relevant, applied, and participatory research landscape in which stroke survivors are involved in research beyond therapeutic rehabilitation practice [36,37]. Participatory health research (PHR), which involves people whose living conditions are the topic of the research as equal research partners in the research process, is becoming increasingly important in Germany [38]. At the same time, the use of this research approach to people with disabilities is still in its infancy in German health research [39].

Photovoice [40] is a visual methodology that is frequently used in qualitative and, above all, participatory health research (PHR) [41,42,43,44]. The number of international photovoice studies involving people after brain injury and especially after stroke has increased in the last 10 years [18,45,46,47,48,49,50]. Although the produced personal photos represent a form of communication that does not require reading or writing [40] and thus provide an alternative to language-based methods in research for people with post-stroke aphasia that shows promising results [45], researchers often exclude stroke survivors with aphasia from photovoice research [51]. The authors of this recent scoping review, therefore, concluded that further studies on the adaptation of photovoice for these groups are needed to facilitate their inclusion in future participatory action research “in ensuring that all post-stroke stakeholders are involved in projects related to social justice and policy for stroke survivors” (ibid., p. 219). Furthermore, the authors do not describe the participation of severe stroke survivors in these studies. Likewise, no photovoice study focuses on the experiences of CM using a PWC in a metropolitan area.

Therefore, the purpose of this study was to involve persons with severe acquired brain injury as research partners in a participatory research process to learn about their everyday life with a PWC in Berlin. This study specifically explored CM with a PWC in a metropolitan area using Berlin as an example to understand community mobility, identify needs for change, and draw attention to persons with severe acquired brain injury using a PWC. Based on these considerations, the preliminary research question for this participatory photovoice study, drafted prior to the start of the participatory project, was: How do people with severe acquired brain injury experience their community mobility in a power wheelchair in the metropolitan area of Berlin, and what changes do they want to initiate?

## 2. Materials and Methods

This study is situated within the context of participatory health research (PHR) [52,53]. In PHR, participation is the guiding intention throughout the research process. Maximum participation should be achieved in the sense of the decision-making power of all people who are actively involved in the research. Collaboration among all participants in the research team is organized in a spirit of partnership, and power relations are continuously reflected. The involvement of stroke survivors as co-researchers aimed to facilitate shared learning and empowerment processes, gain new local knowledge, and initiate change (ibid.). The term “co-researcher” instead of “participants” is used throughout this paper to emphasize the particularly active role of stroke survivors in this research process, beyond the usual form of participation [54,55]. The form and extent of participation in participatory research often represent a continuum; while full participation can sometimes not be achieved, the highest level of quality in community participation in the research process is sought [55]. This means that researchers try to “develop egalitarian partnerships with community members that equalize the decision-making power between researchers and community members … and will make joint decisions that reflect our shared goals and interest in the research project” (ibid, p. 2131). This study was initiated by the first author (TB) as part of her final master’s thesis in the master’s program in Occupational Therapy at the University of Applied Sciences and Arts (HAWK) in Hildesheim, Germany (2016–2018) and was continued after she graduated.

### 2.1. Ethics

The study was approved by the University of Applied Sciences and Arts Commission for Research Ethics (HAWK Hildesheim/Germany, 3 April 2018). In participatory research, research ethics issues have a high priority, as the research often takes place with marginalized or vulnerable groups, and a trusting, equal collaboration takes place over a longer period of time [54,56]. Using photovoice requires reflection on specific ethical concerns, such as the possible violation of privacy by making individuals identifiable and public, the researcher’s influence on the photo’s topics, the photo selection for exhibitions and publications, and questions of photo ownership [44,57,58]. A key critical question is “whose voice” is made visible through the presentation of the photos and titles or captions [44,58]. In this study, the recommendations of the International Collaboration for Participatory Health Research (ICHPR) [52,59] were considered, which demand, among other things, that the ownership of Participatory Health Research Projects “lies in the hands of the group conducting the study” [52] (p. 10). Therefore, the research team collectively decides how best to report and publish the research findings to achieve the group’s stated goals. This refers to all publications (e.g., exhibitions, congresses, articles). The rights to photos and narratives generated in the study belong to the individual co-researchers and are obtained separately for each publication. For this publication, all co-researchers were involved in the selection of the photos, gave their written consent about the use of the photos, and wanted to be mentioned by their first or full names in the acknowledgments.

All participating co-researchers were able to provide written informed consent. Some have legal guardianship for certain areas of life but still have full legal capacity. To account for the co-researchers cognitive and communication impairments, consent was understood as a process, and alternative consent procedures were considered [60,61]. All information about the study (e.g., invitation letter, consent form) was written in easy language [62,63], checked by two stroke survivor peers beforehand, given multimodal, and explained at multiple points in time.

In addition to discussing research ethics with co-researchers during the project, the first author (TB) and her colleague (JK) completed a postscript separately after each meeting for ongoing reflection on research ethics and exchanged ideas regularly. In addition, the first author kept a research journal and used peer discussion with other participatory researchers through an ICHPR training course at the Catholic University of Applied Sciences in Berlin (2018). The critical self-reflections about relationship building and power dynamics in participatory research projects of four research projects from this course, including this study, resulted in a collaborative article [64].

### 2.2. Recruitment of Co-Researchers

People who had undergone specialized, long-term rehabilitation for adults with an acquired brain injury in the past five years and lived in Berlin were invited to be co-researchers in this study. At the time of the study, the first author (TB) and her colleague (JK) were working as occupational therapists in the rehabilitation center and thus had the opportunity to involve these “seldom heard” [65] (p. 163) individuals in the research project.

Recruitment took place in several steps and through different access ways. Invitations were sent to persons who fulfilled the following inclusion criteria:stroke or traumatic brain injury (acquired after the age of 18);acute event at least 1 year ago;motor, cognitive, and/or communication impairments as a result of the acute event,living in an outpatient living arrangement in Berlin for at least 6 months (with or without assistance);provided with a power wheelchair (PWC) during rehabilitation and using a PWC outside the home;interest and willingness to actively participate in a research project that includes several group meetings;enjoyment of photography.

The number of participants was limited to a maximum of five in order to ensure that all co-researchers would have sufficient opportunities to participate in the group meetings [45,50]. People with whom the research initiator already had a relationship of trust were deliberately invited. TB considers herself a gatekeeper to this marginalized group of people [54] because she regularly worked with individuals in this group in a therapeutic context over a longer period of time and was, thus, a part of their lives for a certain period of time (from 1 to 4 years).

When individuals expressed interest in participating in the research project as co-researchers (via email or text message), an appointment was made for a home visit. TB explained her project idea and answered questions and concerns that potential participants might have had. Custom-made information sheets on participatory research and photovoice were used to clarify potential concerns, inspired by a photovoice study with people with intellectual disabilities [66]. Upon written informed consent, a brief questionnaire on demographic data (e.g., living and working situation) and mobility (e.g., assistive device use inside and outside the home) was filled out together. Furthermore, individual time capacities were noted to be able to plan the meetings of the final research team. Likewise, support needs for meeting attendance and desired communication pathways were discussed and recorded. The home visits took place in April 2018 and lasted between 90 and 120 min.

### 2.3. Photovoice Process

The study used the participatory visual research methodology photovoice [44] based on the principles of Wang and Burris [40] “by which people can identify, represent, and enhance their community through a specific photographic technique” (p. 369). People from an often marginalized group are invited to take photos of their living environment under one or more questions, share and discuss them with each other in the group, and make them available to others. The three main goals of using photovoice in participatory research are to empower people to document and reflect on their community’s strengths and concerns, to foster critical dialogue and knowledge about important issues through group discussions about the photos, and to reach decision-makers, such as policymakers and important stakeholders, in the process [40,54].

The photovoice process was guided by von Unger’s seven phases [54] and occurred between June 2017 and May 2019 (Figure 1). The first steps of Phase 1 (Planning and Preparation) were carried out by the first author alone due to the pre-determined time of her master’s thesis. Further preparatory steps of Phase 1 were carried out together with the co-researchers in the first group meeting: getting to know each other, introducing and discussing the participatory research approach, and setting the research question and common goals (Table 1). The following phases involved the co-researchers as equal partners. The research topic CM in a PWC evolved from a literature review and TB’s professional experience as an occupational therapist. A focus group preceding the study with different participants (except one person) who also had a severe acquired brain injury (October 2017) confirmed the importance of the research topic and provided initial foci and questions. Based on this, the first author decided to use a photovoice methodology to provide an appropriate opportunity for stroke survivors to give a visible voice to their concerns in a participatory research approach. TB and JK pre-planned and facilitated the five group meetings.

Based on the co-researchers’ request, the meetings took place in a centrally located, accessible neighborhood center as well as in the former rehabilitation center. Each meeting lasted between 3 and 3.5 h and was recorded with audio and video. Ongoing process consent for recording was obtained in accordance with a dialogic process about research participation at the beginning of each meeting. The video recordings were used to support the transcription since the audio recordings were not always understandable or comprehensible due to the impairments in speech production and manner of speaking. Due to limited resources, a partial transcription of all meetings took place, whereby the verbatim transcription was limited to central statements and dialogues according to the research question of the project.

Co-researchers’ training (Phase 2) included an introduction to photovoice (formulating a storytelling prompt, training on legal and ethical issues, sharing about photos, and desired need for help) using the graphic “A Photovoice Path” by Lorenz [67,68]. The research team decided against restricting the subject of the study (only limitations or possibilities), as it was initially planned by the first author. Instead, a storytelling prompt, “What would you like to tell other people about your life in a power wheelchair?” was formulated to guide the process of photo taking. Co-researchers expressed skepticism about the degree to which change could be initiated, as the original research question suggested. The setting of collective research goals clarified this point; at the same time, it was stated that the first goal of the project was to inform other people (Table 1). The group jointly decided that each person would bring 5 to 10 photos to the second group meeting.

For photo shooting (Phase 3), the co-researchers mainly used their smartphones, and one person was lent a camera as hers broke down in the process of the study. Four of the five co-researchers asked for and received support from the first author in taking photos. They felt uncertain about what they could photograph, were unsure about handling their smartphones due to hemiparesis, or could only leave their homes in power wheelchairs with help from others due to the lack of a grab bar for the transfer (one person). Requested support was given at one to three appointments, during which photos were only taken at the suggestion of the co-researchers. The first author expressed her own opinions only when asked. She then explained the research question and the storytelling prompt again and reported what the co-researchers had already mentioned in previous conversations.

The discussion (Phase 4) and participatory analysis (Phase 5) of the photos occurred in the second and third group meetings. Here, all photos were presented, discussed, and analyzed according to Wang and Burris’ three-stage process: (1) selection of the most relevant photos, (2) contextualization, and (3) codification [40,54]. For stage 1 and 2, the co-researchers presented and described each of their photos. The first author (TB) moderated and facilitated the discussion, and her colleague (JK) noted keywords on moderation cards. In the next step, noted keywords and topics were validated by communication. The research group searched for commonalities, patterns, and differences as well as for “umbrella terms” that grouped ideas and themes together. With verbal direction from the co-researcher, the moderation cards were ordered on a flipchart (stage 3). The resulting categories were discussed, and everyone was invited to assign their photos to the themes or to create new categories. During the discussion, the co-researchers gave working titles to each of their photos. In the fourth group meeting and in subsequent individual meetings, the co-researchers were assisted in formulating their stories to clarify the content of the photos. Wang and Burris describe this under the acronym “VOICE—*v*oicing *o*ur *i*ndividual and *c*ollective *e*xperience” [40] (p. 381). The results of these discussion and analysis meetings were written up and sent to everyone to review in order to prevent possible misunderstandings that could arise, for example, due to memory deficits.

In the third and fourth group meetings, dissemination of the research results were discussed (Phase 6, Presentation and Use). The planning of a first exhibition and further possible publications took place (see Section 2.1 Ethics). The co-researchers discussed the advantages and disadvantages of different exhibition places. Finally, a decision was made that the first exhibition would occur in the former rehabilitation center. This was to meet the co-researchers’ collective goal that, above all, other persons with disabilities need to be informed about the possibilities of a power wheelchair to give courage and hope to those in rehabilitation. The group searched for a title that best expressed their results and could be used for their presentation. One co-researcher’s suggestion met agreement from all: “Back into life—with a power wheelchair”.

In the last group meeting, the project was evaluated by the research group (Phase 7, Evaluation). The following questions were used for reflection: “How did I feel about the project?”, “What did I like?”, “What did I not like?”, “How can it continue?”. All co-researchers received a photo book with their photos and photos from the first exhibition’s opening.

## 3. Results

### 3.1. Co-Researcher Characteristics

Five stroke survivors participated in the research group. They were aged 36–54 years and had undergone long-term inpatient rehabilitation as a result of a first stroke between 2011 and 2017. All five have severe acquired brain injury with a degree of disability (GdB) varying between 90–100% (max. 100%) based on hemiparesis and cognitive and communication impairments. While communication impairments are cited in their medical reports, specific references to cognitive impairments are lacking (Table 2). According to the evaluation of TB and her colleague, JK, as occupational therapists, cognitive impairments in individually varying degrees exist in the areas of attention, memory, and executive functions, including deficits in self-awareness. During the long-term inpatient rehabilitation, all co-researchers were provided with a power wheelchair (PWC). They live in their own apartments and receive support from assisted independent living (Betreutes Einzelwohnen, BEW) or in an assisted living community; one of them is living with intensive support (Wohnen mit Intensivbetreuung, WMI).

The five co-researchers accomplish getting around in their homes differently. The majority use a wheelchair or PWC for independent mobility, one of these persons uses a rollator (walking frame), but rarely, and one person walks independently without an assistive device. All five use their PWC in community mobility, but not in all situations. Four of the five also temporarily use their manual wheelchair. Reasons for this are primarily space constraints, for example, in medical offices, stores, restaurants, event spaces, public toilets, or their workplace. Other reasons include the absence of the PWC due to repairs, as well as issues in the use of the special transportation service (Sonderfahrdienst, SFD). One person describes anxiety in safely managing the ramp when entering/exiting SFD with their PWC. Backing out causes the person anxiety, so all trips are made only with the manual wheelchair. Another co-researcher already walks short distances in her living environment and at work without an assistive device. Thus, for work, the PWC deliberately stays at home, but also because the commute to work is guaranteed with door-to-door special transport service. For all co-researchers, the frequency of leaving the apartment is related to the occupational status and the need for help. The three persons who are employed leave their homes daily. The other two leave their homes one to two days or three to five days/week. The first one is dependent on assistance from other persons for activities outside the home because she needs help to transfer to the PWC due to a missing grab bar in the apartment. The grab bar was prescribed before leaving the rehabilitation facility, but after 6 months, it has still not been installed. As she lives alone, she can only leave her home with her PWC when care services or family visit and help her.

### 3.2. Back into Life—With a Power Wheelchair—Themes from Photos

Six central themes were identified and named by the co-researchers in the group discussion meetings: 1. Built environment, 2. Personal freedom, 3. Me and my power wheelchair, 4. Demands on users of a power wheelchair, 5. Demands on other persons, and 6. Desires for change. The wording of the categories presented is the result of participatory analysis by the entire research team. Discussions with the five stroke survivors made it evident that foreign words and specialized terminology would reduce comprehensibility. Therefore, formulations in easy language were deliberately chosen. Photos and quotes from the co-researchers are included using pseudonyms and referring to the meetings (M1, M2, etc.).

#### 3.2.1. Theme 1: Built Environment

In their photos, the co-researchers depicted different conditions in public spaces or buildings that constrict or, sometimes, enable mobility in a PWC. Uneven boardwalks or streets (such as cobbled streets, see Figure 1), high or steep curbs, and cars parked on boardwalks or crosswalks hinder group members’ community mobility. As a result, they cannot always cross the streets where they would like to, or they swerve their PWC into the street to drive (Figure 2. On the road with the PWC):


*“Yes, … there are pedestrian paths. But they are relatively narrow, everything is relatively green. That means you always have to circle back and forth. And in between there are always these (.) “Huckelpflaster” (cobblestones). When there are driveways.”*
(Charlie, M2).

**Figure 1 ijerph-19-10465-f001:**
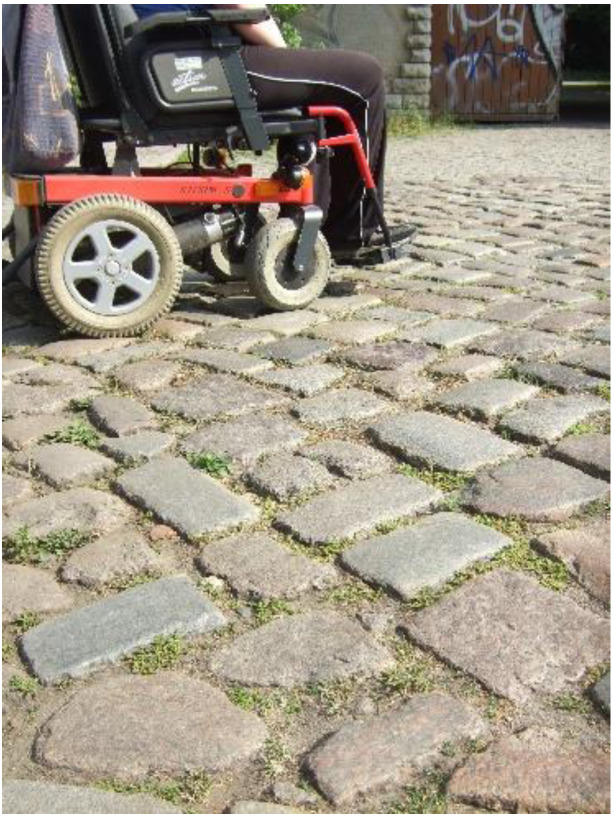
Cobblestones.

**Figure 2 ijerph-19-10465-f002:**
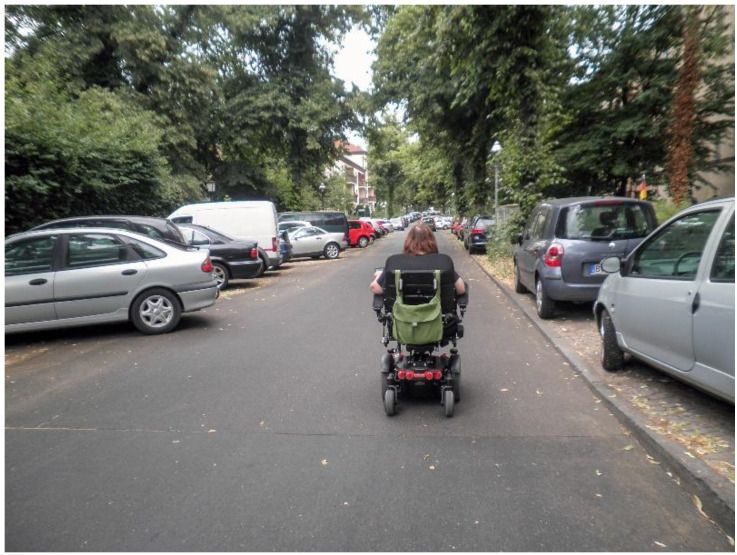
On the road with the PWC.

To board or deboard public transport such as the bus, tramway, underground, and metropolitan railway with a wheelchair in Berlin mostly requires the assistance of the drivers. Wheelchair users must stand at specific assigned places and make their ride requests visible when the train or bus arrives, which feels annoying to the co-researchers. On the other hand, the use of public transport enables various meaningful occupations (Figure 3. Bus stop):


*“I usually go (.) when I (.) visit friends or (.) go for a walk. Or logo (.) [speech and language therapy] do. Yes.”*
(Mika, M2).

**Figure 3 ijerph-19-10465-f003:**
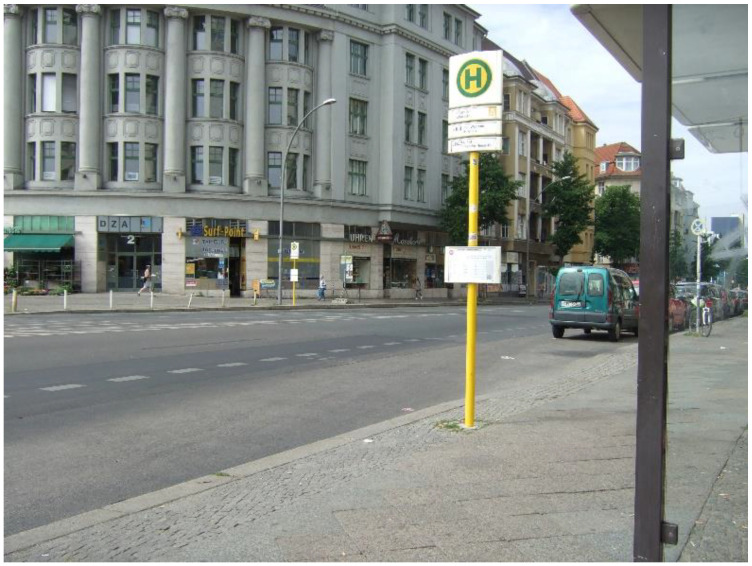
Bus stop.

Some photos showed elevators. Elevators are principally considered useful by the co-researchers, but there is seldom any information about alternatives if they are out of order, and not every station in Berlin has an elevator yet. Restricted accessibility and deficient space for power wheelchairs in buildings are also seen as a barrier to doing what one needs or wants, especially in medical practices. Stairs often prevent people from choosing a family doctor close to home—unless one can walk a few steps again, as visible in the photos from one co-researcher (Figure 4. Medical practice).

#### 3.2.2. Theme 2: Personal Freedom

Four of the five co-researchers narrate the possibility of independently doing various meaningful occupations outside their homes, such as going to the cinema, cafés, parks, or for a walk in the neighborhood using their PWC. The PWC provides an important resource in community mobility (CM), for example, by facilitating leisure activities. Lukas reports on his visits to the cinema with a friend who also took the photo of him (Figure 5. Cinema):


*“*
*You can see the cinema there. Alexanderplatz. And I did that because I’m there quite often. (.) … Because all the cinemas there are barrier-free. … In the PWC or wheelchair you sit in the back. And you have a good view of the screen. That’s why I think it’s so good.”*
(Lukas, M3).

**Figure 5 ijerph-19-10465-f005:**
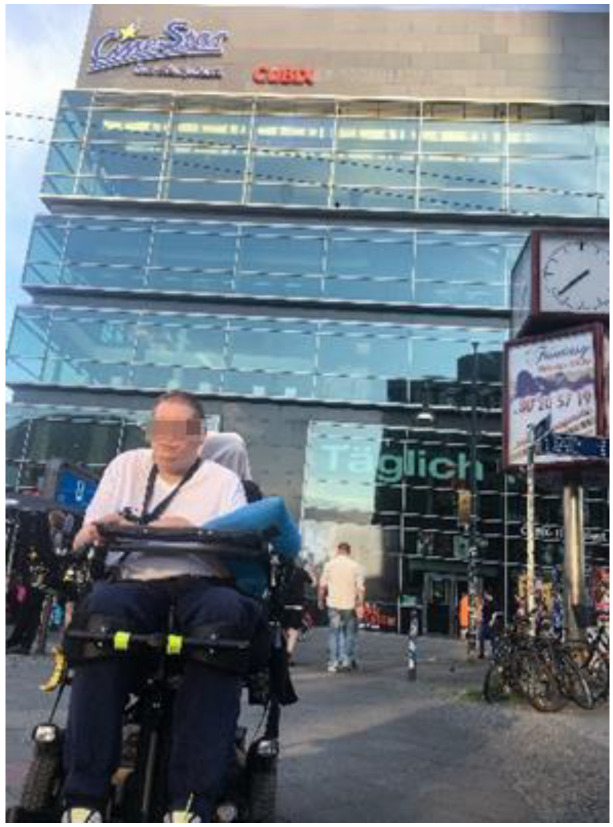
Cinema.

For another person, it is important to simply be in a public space, to be among and part of other people:


*“I enjoy that (.) when I (.) go to market hall. And (.) then (.) people arrive. And watch (moves her head to do so). And (.) people approach me. “What did you do” and dedede (.) … But (.) this is (..) my home (gestures with her hand). (.) Yes. (.) Like this.”*
(Mika, M2).

With the support of the first author, Alex took a photo in a public park near her home, which she likes to visit regularly. She describes how the PWC compensates for her limited walking mobility, enabling her to resume individual interests and routines, thus supporting her self-determination and well-being. Alex formulated for her photo (Figure 6. Schlosspark):


*“I like to be outside with my PWC. I like to go to the Schlosspark. I love the trees—always have. I like to be in nature. I can’t get there on foot. With my PWC, I can go there by myself when I want. My wish: Despite my physical limitations, I would like to do the things I like to do. And that do me good. The PWC helps me to do that.”*


**Figure 6 ijerph-19-10465-f006:**
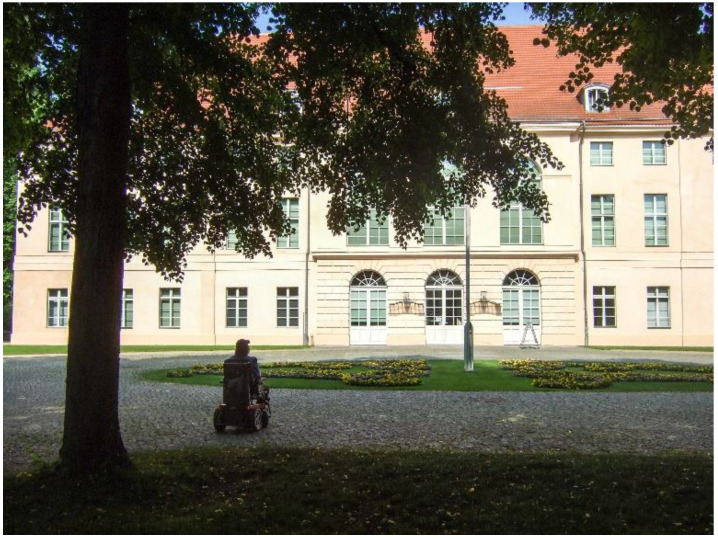
Schlosspark.

#### 3.2.3. Theme 3: Me and My Power Wheelchair (PWC)

The central role of the PWC in enabling meaningful occupations and choice causes a high identification with the assistive device. One member of the research group named their PWC with her second surname (Figure 7. Bonita):


*“Uh (runs her hand over the armrest of her PWC and considers) (…) Bonita. And (..) I love (..) (taps hand on armrest in time to speaking) PWC.”*
(Mika, M2).

**Figure 7 ijerph-19-10465-f007:**
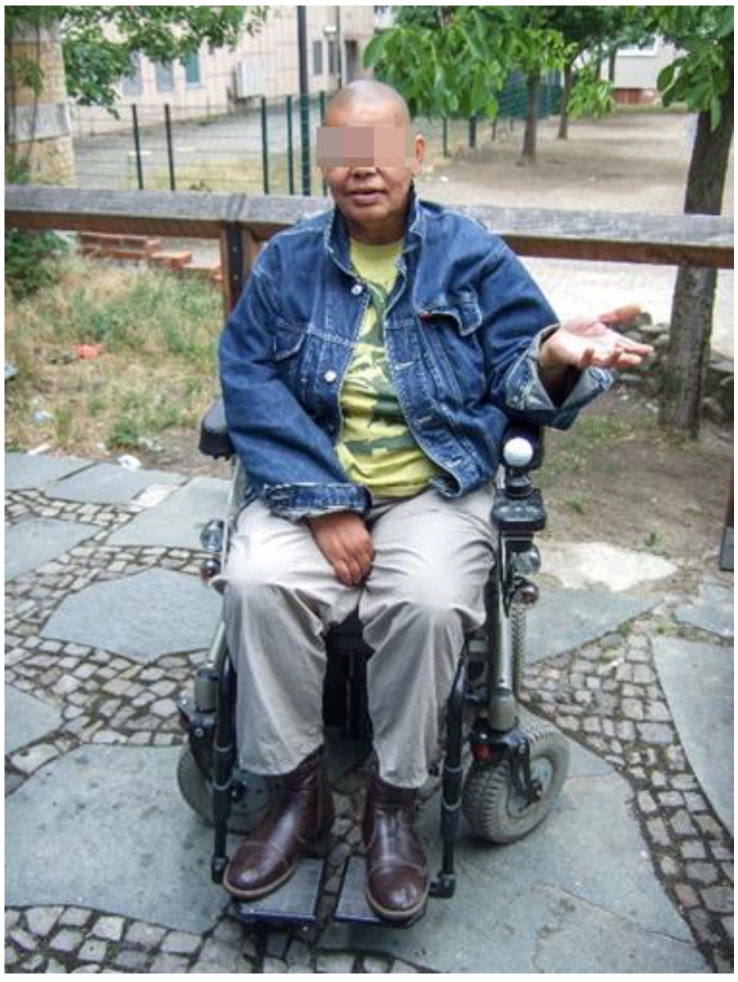
Bonita.

In the meetings, she used this name repeatedly when she spoke praisingly of her PWC. She is very satisfied with her aid because it enables her to do a lot: “*My wheelchair is (.) Bonita. (.) And (.) love freedom. (..). Yes.”* (Mika, M3).

These very personal descriptions led to a joint discussion in the group on how this could be represented in a photo. One co-researcher expressed the suggestion that Mika should take a picture of herself with her PWC. When asked where this photo could be created, she responded, “now, here”. For the making of the photo, the environment was not important because the photo should primarily depict a satisfied Mika with her assistive device.

Another co-researcher even got a license plate with the words “*se-xy 50*” for her birthday and is thinking about attaching it to her PWC as she did with her former car. She has not attached the self-made license plate to her PWC yet, but can imagine doing so in the future if her relationship with her PWC evolves:


*“I … have always given my car a name. So, I think I would then also, if I can identify with that, somehow. … If I can then say that it is mine. (.) Then I would also give it a name.”*
(Charlie, M2).

Another co-researcher associates his PWC with a tank (Figure 8. T34/Tank). This association refers to both the size and the driving characteristics, which are either helpful or a barrier, depending on the context. The PWC is *“(.) also designed to (.) simply take the terrain”* (Chris, M3). In confined spaces, such as a café, this could lead to unwanted situations:


*“If I … drive into the café (.) too briskly (.) … then I drive into the tables. (.) A path of destruction in”*
(Chris, M2).

**Figure 8 ijerph-19-10465-f008:**
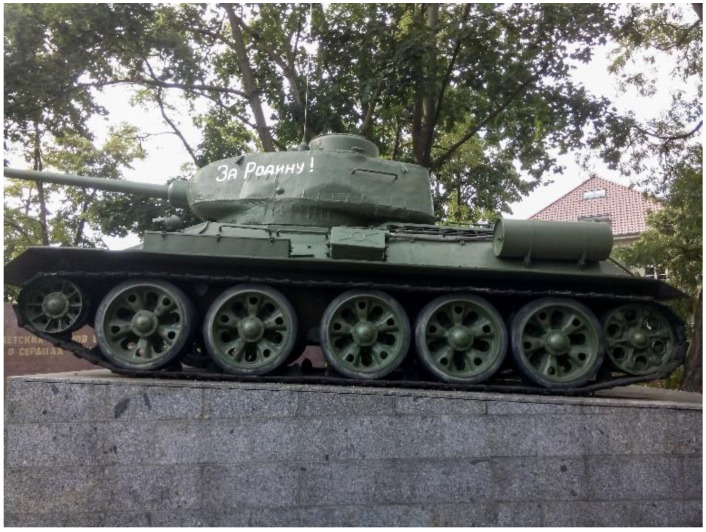
T34/Tank.

Despite the off-road capability of his PWC, there are also *“insurmountable obstacles*” for him, such as curbs that were too high: *“Well. And (.) sometime is also times closing time. No?”* (Chris, M2).

#### 3.2.4. Theme 4: Demands on Users of a Power Wheelchair (PWC)

Independent mobility with a PWC needs planning and preparation. Co-researchers describe that they have to be informed beforehand about the accessibility of, for example, a cinema and how it is necessary to check if there are elevators at the transport station when going to the cinema. If they want to use a special transportation service, they also have to book in advance. Before longer rides with a PWC, the battery needs to be checked. PWC users have to know alternatives if plans need to be adapted, they have to be able to ask for assistance, and they have to have knowledge of how to get assistance. A co-researcher describes that it is not always known where one can find or receive appropriate help, for example, who could accompany one on a voluntary basis, or where to find reliable information about accessible places: *“Yes, but it’s not always (.) clear. So, barrier-free restaurants. That is totally difficult to find!”* (Charlie, M3).

In addition to planning and preparation, it is also important to know alternatives, have knowledge about the PCW, as well as to be able to ask for help. Broken elevators sometimes cause a change in the original route. If the PWC is broken, one has to know how to unblock it and ask people to push. It is also necessary to know a service or emergency number as well as one’s rights. A co-researcher described that while his PWC was being repaired, he was not provided with a replacement PWC despite being entitled to one. Upon inquiry, he was told that no PWC was available in his size. As a result, he was unable to do his grocery shopping and lived on his supplies, as he reported after a meeting. From this account, it is clear that knowing one’s rights and being able to advocate for them to others is also a requirement—and to ask for help if one’s rights are not met. A co-researcher concludes: *“And then I think you really have to be well-structured.”* (Charlie, M3).

#### 3.2.5. Theme 5: Demands on Other Persons

The co-researchers described many situations where they missed appropriate assistance and support from other persons. A shop assistant coming out of a small shop with stairs by his initiative and asking what one needs is welcome. In another situation, one person was offered help by the waitress at a café because the toilet was not barrier-free. She describes very impressively how this was perceived as ‘crossing the line’:


*“Sure, with my brother I would manage that. (.) Well, I would also be able to walk 2, 3 steps (.) to the toilet. That would not be a problem. (.) But I wouldn’t do that with a complete stranger. … (laughs). I don’t say to the waitress: “Yes, go with me to the toilet!”*
(Charlie, M3).

The co-researchers also describe exclusions and prejudices they experience in everyday life. Lukas described how, when visiting a restaurant with family or friends, he was the only person not offered a menu on several occasions. His conclusion is: “*Well, if you’re in a wheelchair, you’re automatically stupid*.” (Lukas, M3).

The bus drivers of public transport were often described as verbally or non-verbally unkind for reasons that are not clear. *“Uh (.) grumble”.* (Mika, M2) *“One can see it in the face.”* (Charlie, M2).

A co-researcher described an experience: *“I had that last time, too. (.) I was on the road with …. And we were trying to catch the M … [number of the bus] in the direction of …. She ran ahead. Because she already saw that he was coming. (..) And then I ran after him. And then he was in a really bad mood and threw his gloves and everything he had (..) onto this shelf in front. And then (.) asked in all seriousness why we were going with him and not taking the later bus. (.) The next one is coming soon. … Because then they don’t have the will/they don’t have the will to make the ramp out.”* (Charlie, M2).

The research group thought about how this could be expressed in a photo. One co-researcher spontaneously stuck out her middle finger and received approval from the other co-researchers. As a result, Figure 9 was created as a joint photomontage (Figure 9. Stinky finger at a bus stop).

Another important theme was the adequate individual provision of assistive devices in relation to the context. One group member was able to go shopping independently with her PWC while in rehabilitation. Independent transfer to her PWC is not possible in her own flat because for nearly one year, the supplier of medical equipment did not deliver a grip bar and other adaptations for her PWC (Figure 10. Power wheelchair).


*“That it is not enough (.) to have such an expensive part [PWC] there. But that you also have to get it adequately adapted. … Yes. And now I just sit in the shack. (.) It’s really like that.”*
(Charlie, M2).

**Figure 10 ijerph-19-10465-f010:**
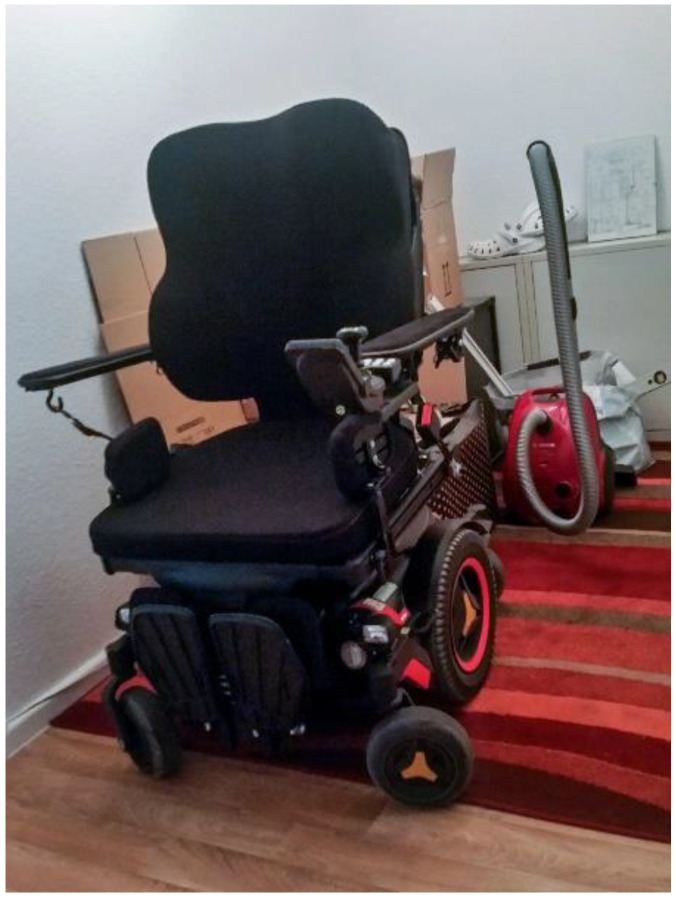
Power wheelchair.

#### 3.2.6. Theme 6: Desires for Change

The co-researchers constantly—in all group meetings—expressed their wishes for change, such as more accessible medical practices or stores. *“Well, I think that (.) not accessible stores (.) too few for us. … where you can drive in and look. (.) Like a normal customer. (.) Customer is king.”* (Charlie, M3).

Basically, they all wish to be independently mobile in buildings and public transport with their PWC—without the assistance of other people, such as being dependent on someone who handles a ramp. The co-researchers expressed their wish for recognition and appreciation of their knowledge and experience and to be treated with an attitude of equality. They assume that people are not aware that the co-researchers had a “normal” life some time ago.


*“I was healthy until five years ago, too. And jumped around. And now I am sitting in one of these. I imagined it differently, too. And tomorrow you could be hit by a stroke!”*
(Charlie, M3).

In the provisioning process of assistive devices, stroke survivors want to be heard more so that the assistive devices meet their individual wants and needs. Likewise, they would like more support in advocating for their rights with payers, such as insurance. One co-researcher reported that she spent a lot of time making phone calls inquiring about approval or delivery. She expressed during the first home visit, “*I could spend my free time in other ways!*”

A central concern of the research group was and is to communicate courage and the possibilities of assistive devices like PWCs to other persons with disabilities.


*“This is the place [rehabilitation center] where you are made fit again… for life (.) And (.) … I’m sure some people have already resigned. (..) There’s no progress at all. And (..) And yes, everything sucks. And so on. (.) But (.) the people should also see that it (.) goes up. That there are assistive devices.”*
(Chris, M3).

### 3.3. Dissemination of Findings

In January 2019, the findings of the study were exhibited for the first time in the cafeteria of the long-term rehabilitation center over four weeks. The place was decided upon the majority request of the co-researchers. Current rehabilitants were present, in addition to staff and management of the institution. After introductory, appreciative words from the head physician, the co-researchers and the first author presented the project, the results, and their concerns (see Figure 11). The representative for seniors and passengers with disabilities of the Berlin transport company (BVG) was invited to the exhibition opening and showed interest in the results. However, as often, the initial contact that was established at the exhibition was not followed up.

Afterward, three co-researchers expressed a general interest in accompanying the authors to congresses and reporting on the project—at least if these were to take place in Berlin. One co-researcher went to the opening of the exhibition at the Catholic University of Applied Sciences Berlin (conference of the Berlin Workshop for Participatory Research, March 2019), and one co-researcher and another rehabilitant reported about the project at the opening of the exhibition at an occupational therapy school in Berlin (December 2019). Furthermore, the results were presented at four scientific conferences in the form of verbal presentations, at two of which the first author was accompanied by a co-researcher.

### 3.4. Evaluation

The planned evaluation after the first exhibition in early 2019 had to be canceled due to illness and was postponed by three months. All co-researchers spoke positively about the project that had been carried out. At the same time, most of them found it difficult to make an evaluation and thus a conscious reflection. Two smileys cards (sad and laughing facial expressions) supported the process: the two cards were attached to the flipchart to visualize the questions, “What did I like?” and “What did I not like?”. Two co-researchers emphasized the importance of the exhibition in the rehabilitation center, and one co-researcher proposed the idea of meeting other, former, and familiar rehabilitants again and doing something together. All emphasized sharing with each other and how they experienced the associated feeling of not being alone with the problems they experienced. One co-researcher expressed the amount of time and organizational effort associated with the meetings as negative. Above all, the long journey of one hour by SFD was perceived as burdensome. In order to share their own motivation for the project and to communicate it to others, e.g., at congress presentations, the authors suggested that short statements by the co-researchers should be recorded on video. Four agreed to this idea, and the fifth person did not want to do this.

## 4. Discussion

The aim of the presented participatory study was to find out how people with a severe stroke experience their community mobility (CM) using a power wheelchair (PWC) in the metropolitan area of Berlin and what changes they want to initiate. Our results show that all co-researchers regularly use their PWC outside the home despite numerous existing barriers and challenges experienced on a daily basis. Thus, the assistive device prescribed during rehabilitation occupies a central, if not irreplaceable, part in the independent lifestyle of the co-researchers. The impressive descriptions and discussions of the co-researchers reveal that successful CM with a PWC is a very complex construct; it depends on several factors and has an individual level of meaning. In the following section, we want to discuss the most important results and methodological considerations for future research before we finally mention the limitations of the study.

### 4.1. Complexity of Community Mobility Using a Power Wheelchair

#### 4.1.1. Accessible Environment and Transport Possibilities

The results of the presented study support the statement that CM is a complex construct for social participation that depends on many factors [8,71,72], especially for people after acquired brain injury who are mobile in a wheelchair [20,24,73,74]. The most common physical barriers in the environment described by our co-researchers and previous studies were access to buildings and the surface condition of footpaths (potholes, curbs). This shows that public space still does not provide equal access to all people and that wheelchair users are still not considered enough in inclusive urban planning in Germany. A recent statement by the German Institute for Human Rights (Deutsches Institut für Menschenrechte, DIMR), which was commissioned as an independent national human rights institution for the implementation of the UN CRPD, confirms that the strategic planning of the administrations in Berlin hardly or not at all take the special mobility needs of people with disabilities into account [75]. This statement has wide-ranging, public health-relevant consequences for the persons concerned. For example, due to only limited barrier-free public transport, lack of barrier-free cabs, and complicated billing modalities for the special transport service (SFD), many people had great difficulties attending their SARS-CoV-2 vaccination appointments in the established vaccination centers in Berlin in 2021 (ibid.). Our co-researchers also describe the limited accessibility and accessible equipment of medical practices: only one in three medical practices in Germany is barrier-free to date [76]. Our results reinforce previous research from other countries that people with disabilities often experience health disparities resulting from the inaccessibility of healthcare services, see, e.g., [77]. To promote the social participation of wheelchair users in Berlin, there is a need for the coordinated overall planning of public space, including footpaths, roads, buildings, and means of transport, as well as a general national legal obligation for accessibility in all areas of life, as has been implemented in other countries, such as Austria or the U.S., for decades. It is important to systematically involve the people concerned in these decisions. It should also be discussed and investigated to what extent the SFD can be reduced by expanding generally accessible transport services, such as barrier-free call cabs [75]. This could not only contribute to a reduction in costs but, above all, realize full and equal participation in the sense of the UN-CRPD and minimize the dependence on others, such as family members.

Previous international studies show that people after stroke are more likely to use special transport services or be passengers in cars with family members [15,26,78], but partly without their PWC because transporting it is not possible in most cars, so the PWC then remains at home. The experiences of the co-researchers in our study do not coincide with these previous findings: only one person used SFD regularly, although all were eligible. The low use of special transportation services like the SFD in Berlin was explained by the co-researchers as a lack of flexibility and unreliability of the service, as shown in a previous study from Iceland [79]. Desired trips have to be planned and booked with a lot of advance time and organization, the SFD can only be used at certain times of the day, and sometimes there are significant delays. Their descriptions illustrate that SFD does not automatically increase CM since it does often not support spontaneous CM and that other transport options should.

#### 4.1.2. Invisible and Unexpected Challenging Experiences

The explanations on the use of the SFD and further descriptions of the co-researchers reveal, in addition to an accessible environment and transport possibilities, that it requires much more to be independently and self-determinedly mobile with a PWC in the community. First, it requires a massive amount of planning, which is not visible to other people and, thus, often not known. This highlights that CM does not start when leaving home, but much earlier. PWC users must obtain a lot of information before making their daily trips to reach their destinations on time and use the services provided there. Thus, they need comprehensive strategies of where to look for this information and access to the skills to use information sources, such as the internet. A recent scoping review highlights that by elaborating the different cognitive functions required for safe use of powered mobility devices [80]. Similarly, such planning efforts require an enormous amount of time. These barriers to access became highly “visible” to the non-PWC-using researchers as they attempted the collective dissemination of the research findings. The participation of two co-researchers in a scientific congress in a city 500 km away required numerous planning steps as early as 6 months before the event, including booking and reserving train travel with as few changes as possible and requesting assistance (by a lift) to get on and off the train, researching and booking accessible accommodation that met the co-researchers’ individual needs, and booking an assistant for dressing and undressing. Because of this elaborate planning and the fear that something might go wrong, only a few co-researchers were willing to go on this “adventure”. It took a lot of positive reinforcement for two co-researchers to get involved. The joint trip almost failed at the starting point because the scheduled bus to the train station did not run that morning, and the co-researchers could not change to any other barrier-free public transportation (subway or cab). Fortunately, a wheelchair space was still available on a later train service that day, so the congress visit could be made together. These experiences and findings have practical implications for rehabilitative practice, developers of digital assistance systems, as well as the planning of participatory research projects. In rehabilitation, in addition to training with the PWC in the desired environment in the community, the focus must be on counseling stroke survivors and their caregivers on what planning steps are necessary and where to find relevant information. Peer support groups or peer counseling may be an appropriate intervention. Previous research has shown that these can help stroke survivors cope with their lives [81,82]. Likewise, it is important to educate about the fact that spontaneous, unexpected challenges can occur and to prepare possible solutions together. This includes, for example, always carrying a cell phone and knowing which person to call in a difficult situation, or alternatively, which app can generate alternative barrier-free travel routes using public transportation. This requires digital assistance systems such as apps that can be used without assistance by people with cognitive impairments. When planning participatory research projects, sufficient time and financial resources must be available, as well as the willingness to engage in the adventure of the experiences of the researchers concerned to establish the required equality in the collaboration. In order to achieve not only symbolic or sham participation, it is necessary to adapt the existing funding conditions for participatory health research in Germany [83].

#### 4.1.3. Individually Meaning of Community Mobility

The photo narratives of our co-researchers reinforce the diversified meaning of CM. First, being mobile outside the home enables a variety of activities of daily living, especially for shopping and numerous leisure activities, which is consistent with the findings of other international studies [20,21,24,73]. Second, to be mobile in the community means “‘being a part of” and being “a respected, valued member of the community” [73] (p. 55). The co-researchers describe how they can do things that are personally important to them when they want, for example, visiting a café or a park. CM thus contributes to maintaining their own identity. Alternatively, as Nanninga and colleagues formulated, “mobility is considered as a way to connect places that are meaningful to individuals rather than as movements from A to B” [8] (p. 2016). Third, the photos and narratives of the co-researchers also indicate that the statement “to be mobile” refers to a limited spatial area. This means that, above all, the possibility of being able to carry out everyday activities in one’s own local living environment—in Berlin called “Kiez”—with the PWC without accompaniment. There seems to be an individual mobility radius that is essential for satisfaction, social participation, and belonging: “*That’s enough. … Yes. In my “Kiez”. … And still I am on the road in Berlin*” (Mika). This statement also has practical implications for rehabilitative practice; it reveals the high importance of a locally community-based, socially oriented rehabilitation for stroke survivors in order to enable independent and meaningful CM and social participation [84,85,86]. Further research can investigate how even a limited mobility radius contributes to this group’s independence, self-confidence, and quality of life and to what extent this represents a special feature for the included target group of people with cognitive and communication impairments.

### 4.2. Identification with the Power Wheelchair

Despite the multiplicity of barriers to using a PWC in CM, the co-researchers highlighted the important role of the PWC. They show a high degree of identification with the PWC, which led to individually naming them in some cases. This expresses a kind of personal ‘ownership’ and even empowerment. The PWC seems to be much more than an assistive device object that compensates only a physical impairment. This could mean that the PWC has become part of their subjectivity because it gives back control over their lives and autonomy that they had lost due to the disease. With the PWC, they are once again able to decide for themselves when and where they go. Evans’ findings support that the provision of a PWC can lead to overcoming the experienced occupational deprivation [21] (p. 551). Wilcock defines occupational deprivation as an occupational risk factor, an external circumstance that keeps individuals from performing meaningful occupations and that, if prolonged, poses a health risk [87]. Disability as a social construct is one such influencing factor and applies, for example, to people who are dependent on a wheelchair (ibid.). The relevance in relation to subjective health perception is shown in the statement of Charlie, who currently cannot use her PWC without assistance: “*If my wheelchair (laughs) would work, I would feel better. (.) That’s just a bit on my stomach*” (Charlie).

The core value of the PWC for severe stroke survivors highlights the central importance of the provisioning process of assistive devices in rehabilitation. Indeed, a key finding of our study is that this high level of identification with the PWC was not present from the outset, which might be one reason why the co-researchers highlight the desire to inform peers. All co-researchers described that they were unsure to what extent they could “become friends” with the PWC when being introduced to and testing a PWC in the rehabilitative context. Only through the positive experiences during the use outside the home—in the community—did the high value increasingly become apparent to them. Their own experiences led the co-researchers to the objective of the project “to inform other affected people about the possibilities of a power wheelchair” (see Table 1). Future research should further investigate the aspect of peer support groups or counseling in the context of assistive device provision with a PWC. In addition, all actors (e.g., physicians, allied health professionals, as well as payers) involved in the provisioning process should be aware of the possibilities of PWCs so that they can point them out in counseling and during testing of driving while minimizing existing fears. In Germany, there seems to be limited recognition of these possibilities among payers to date, as evidenced by high rejection rates of around 19–36% for assistive devices such as wheelchairs, PWCs, and electric scooters in the few existing studies [88,89]. The provision of wheelchairs in German health care often focuses only on the close range (“Nahbereich” such as living space) and thus does not meet the participation requirements of persons with disabilities [32]. Especially people with multiple disabilities can achieve an effective improvement in their independently manageable radius of CM by being equipped with a PWC or with an auxiliary drive (ibid.).

### 4.3. Methodological Considerations

Regarding methodological considerations, our study shows that photovoice is an appropriate, accessible method for collaborative, participatory research with people with severe strokes. Even if reading and writing skills are not necessary, the importance of verbal expression skills was evident in describing, discussing, and analyzing the photos. It was helpful that the first author had accompanied some of the co-researchers in photo taking and could support the communication through her memories when needed. In addition, our experiences indicate the benefit of a “mixed group” of stroke survivors with and without communication impairments. The other co-researchers intuitively showed understanding and responded by describing what they saw in the photos and what it might mean. The communication-impaired person confirmed or denied the statements verbally or with head nodding/shaking. However, it remains to be seen whether Mika and Alex, the two persons with aphasia and apraxia of speech, were able to communicate all their concerns. It is also not clear to what extent the verbal statements of the other members of the research team influenced this, according to the key question, “whose voice” is made visible through the photos and narratives [44,58]. When selecting the photos for the exhibitions as well as for this article, the first author noted that she would have chosen different photos in some cases. For example, it was important for Alex to include Figure 1, Cobblestones, in the article. Even though the co-researchers were involved in this step of the publication, it should be noted that they were not involved in the writing of this manuscript and text selection. According to Evans-Agnew and Rosemberg, this reflects a widespread challenge in photovoice research designs [44]. Some selected photos show the individuals in portrait, thereby making them publicly identifiable. Although this aspect and possible associated dangers, such as stigmatization, were addressed with the co-researchers, the consequences of publication can only be predicted to a limited extent. Methodological considerations should also take into account the positioning and reflection of the two researchers (TB, JK) who prepared and conducted the research meetings. The pre-existing trusting relationships from the therapeutic rehabilitation setting were not seen as limiting but rather as a special resource [64,90]. Nevertheless, it required a constant reflection on their roles, expectations, possible influence, and open communication about them with the co-researchers. For example, at the beginning of the research project, it was important to acknowledge that the people included did not see themselves as “co-researchers” and thus did not speak of “our” project until the study was underway. Reflections of an inclusive research team on their working relationships support the experience that it takes shared time and situations before a “we” emerges throughout the team [90]. Future research should investigate further accessible methods that enable participatory research by people with severe communication impairments.

### 4.4. Limitations

There are also some study limitations. First, the co-researchers did not receive cameras, were not provided with special camera training, or assessed for camera adaptations, such as one-handed access, as in other studies [45,47]. This could have reduced the support and a possible influence by external persons when taking photos [51]. Joint “neighborhood walks” through a district of Berlin would probably have been helpful as a first step to take pictures and try out the cameras [66]. In this study, the co-researchers not only had hemiparesis, which made it difficult for them to handle the cameras or smartphones, but they were also on the road with a motorized assistive device, which is controlled by the active hand. In future studies, newer technical possibilities could be tested, such as cameras attached to the head (headwear), which can be used to take hands-free pictures from one’s own perspective. Second, the participatory involvement of the co-researchers could have been higher under different circumstances, e.g., from the beginning of the planning phase. This research project was developed in the context of a master’s thesis and thus had limited resources. Due to this, not all of the co-researchers’ concerns could be implemented in the research process. The majority expressed a preference not to hold the joint meetings at the former rehabilitation center, which is located in the suburbs of the city and thus would have meant a long, exhausting journey for some. The search for a centrally located, barrier-free room that could be used free of charge due to limited financial resources was successful. Since the co-researchers live in different districts of Berlin, travel times remained at 45–60 min for individuals. The length of the meetings was not reduced because of this, although most of the co-researchers showed signs of fatigue despite breaks, which can be discussed as overload due to disregard of the reduced capacity. Third, the co-researchers’ cognitive impairments were neither directly named by them nor cited in their medical reports. Not all co-researchers provided medical reports; this was a voluntary option. Even though the evaluations of cognitive impairments are based on the first author’s and her colleague’s (JK) years of professional experience with this clientele, these evaluations may be liable to bias and might not be relevant to one’s voice being heard, but for searching for alternative methods. Fourth, more public exhibitions in the different districts of Berlin could have increased the impact of the project. Finally, in terms of sustainability of the results and the associated initiation of change, it would have been useful to involve political decision-makers from Berlin transport companies, representatives of health insurance companies, service providers of assistive devices, and medical and therapeutic staff involved in the provision of assistive devices in a follow-up project.

## 5. Conclusions

This participatory photovoice study demonstrates that the active inclusion of severely affected stroke survivors as equal co-researchers in the research process is possible and provides important insights and findings. The power wheelchair plays a crucial role in the lives of stroke survivors involved in this study, as it decisively supports their desire for a self-determined and independent lifestyle by enabling community mobility in Berlin and social participation for them. However, community mobility should be understood as a complex, individualized construct that requires both an accessible environment and multiple planning strategies by PWC users. According to their own statements, after a long period of rehabilitation, these people have “*managed to jump back … into life*” (Chris, M3).

The practical implications of this study lie in the need for increased involvement of the target group in rehabilitation, research, and public planning processes. In rehabilitation, interested individuals should be involved as peer experts in the various stages of the provision process of assistive devices, and access to a power mobility device should be supported by the interdisciplinary rehabilitation team as well as by payers. The special challenges in community mobility should be extended by peer support services. Further research on this issue is needed.

## Data Availability

The datasets used and analyzed in this study are available from the corresponding author on reasonable request.

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
