# Peer review of "“Back into Life—With a Power Wheelchair”: Learning from People with Severe Stroke through a Participatory Photovoice Study in a Metropolitan Area in Germany"

_ijerph, 2022, doi:10.3390/ijerph191710465_

Round 1
Reviewer 1 Report
Reviewer comments
Thank you for giving me the opportunity to review the article „Back into life – with a power wheelchair: Learning from people with severe stroke through a participatory Photovoice study in a metropolitan area in Germany“.
To my opinion this manuscript offers important insights regarding the needs and challenges of wheelchair users that suffered a stroke form their individual perspective. I was impressed about all the efforts and engagement that was necessary to conduct this study with a “true participatory approach”. The study is innovative and appears to be important the audience of the journal. Overall, the methods and results parts were the most matured parts of the current manuscript and I do thank the authors for the “tick” description.
However, the manuscript as it is presented today requires major revision. Currently, the article appears to be too long and not focused enough towards the research question. It would benefit from condensing. I do propose to sharpen introduction, methods, discussion and conclusion part.
Moreover, I do have the impression that the authors’ team does not own extended experiences in scientific writing. Within the entire manuscript weaknesses on how text is referenced to literature does appear. In order to augment its quality, I strongly encourage the authors to read again critically and insert missing references.
In the following, specific optimisation comments are offered along the sections of the manuscript.
Abstract.
· Please clarify who was analysing the results. When applying participatory research, the word discussed appears to be too unclear.
Introduction.
· The introduction needs to be condensed to lead the reader towards the research question. In the moment I have the impression that a lot of text is not cited with the corresponding references (see within the manuscript). Please overwork the text carefully and insert the references.
· It would be helpful for the reader to define what you mean with participatory research and also the photo voice method (e.g. reference).
Methods.
· How was the informed consent given to the study by the participants? Written/verbally/? Can you provide an ethical approval number? I
· It is mentioned that “information about the study (e.g., invitation letter, consent form) was written in easy language” How did you check this? Where there any peers involved in the process? Did you use an instrument?
· Recruitment of Co-Researchers Can you stated how many minutes lasted the home visits?
· 2.2.-2.7 seem to be the phases of the PHR process. Maybe you can consider to rephrase the headings? To me it was misleading.
Results.
· The characteristics of the co-researchers part can be shorten using only text or table. Please insert the table near the information. Are the names used in the quotes pseudonyms?
Discussion.
· In order to follow the argumentation lines, it would reader-friendlier if more subheadings will be utilized. Beside the authors are using the multiple references, the results are not extensively discussed with the international literature. In some passages it appears to be more like an introduction than a discussion. I promote to interconnect the literature with the important results of this study.
· There is an imbalance in discussing benefits and limitations. Moreover, the critical reflection of what should or could be done better is important. Try to condense and balance this part.
· The methodological discussion of using PHR and Photovoice, engaging the end-users as co-researchers may be made more prominent.
Conclusion
· The benefit of using PHR (e.g. instead of qualitative) might be stronger highlighted.
Thank you for conducting this important research and putting our attention towards this “under-researched” issue and involving these “who are marginalized” with an active voice in your research.
Best greetings
Author Response
Dear Reviewer,
we would like to thank for the constructive suggestions that improved the manuscript. We have carried out a major revision in accordance with the reviewers' comments. Any revisions to the manuscript have been marked up using the “Track Changes” function, such that any changes can be easily viewed by the editors and reviewers. The literature references in the deleted text sections have been removed so that the numbering of the sources is correct.
We have provided our point-by-point response to the reviewer's comments and further details on the revisions to the manuscript in a separate file. The line numbering in the separate file refers to the submitted manuscript version. Please see the attachment.
We hope that with the changes made and the explanations in the point-by-point response, the manuscript will be eligible for publication in the Special Issue of the International Journal of Environmental Research and Public Health.
Sincerely yours,
On behalf of all authors,
Tabea Böttger

Reviewer 2 Report
Overall, the manuscript is clearly written. Photovoice is considered an empowering method to engage people in participatory research and social change. This practice can raise ethical dilemmas. I recommend the article of Abma, Breed et al. Whose Voice is it Really? Ethics of Photovoice.. to reflect more on ethical dilemmas.
Line 103: To date, no studies could be found...
I believe this needs some more literature research because there are many recent publications on using participatory action research in community rehabilitation for people with acquired brain injury.
The Dutch organisation ZonMW is supporting a participatory research project on communication and participation for people with aphasia. The project's name is COPACA. Website: www.meedoenmetafasie.nl
Mail: info@meedoenmetafasie.nl
Maybe this can be interesting for you?
I wonder why the participants in this research are called co-researchers. In my view they are more participants. Is it possible to explain more on the relationship, teamwork, working alongside with them as colleagues?
People with acquired brain damage are participating in your meetings and each meeting lasts between 3 and 3,5 hours: I wonder if they were not too exhausted after this meeting? How was this decided and evaluated?
As the authors describe some basic components are lacking: no photography training but I also regret the absence of a social action plan. The community was not involved in this project, am I right? Maybe this project could have had more positive impact on the attitude and mindset of the community if f.i. an exhibition would have been planned in the heart of the community, and not only in the care organisation? Who needs this exhibition? Is it the survivers of stroke who need to be informed only?
273: co-research
Author Response

(The authors gave the same response as above.)

Reviewer 3 Report
Dear Authors,
I spent quite long periods of the time for analyzing this study. This academic paper was classified as belonging to participatory action research (PAR) and also within the category of qualitative research(QR). A paper of this nature focuses on a deep and comprehensive understanding of small-scale cases rather than the statistical procedures or quantification of ordinary quantitative research, and it is important to be well organized and planned because it contains many advantages and disadvantages at the same time.
In this sense, as participatory practice research, the “photovoice” research method, a qualitative research method, expresses one's values or thoughts through 'photography', studies perception in depth, and could share emotions and thoughts that cannot be expressed through language alone.
First of all, I was surprised that there are still barriers related to electric wheelchairs for the disabled in developed countries where the city system has been well-organized in Germany.
In my opinion, this paper faithfully implements research ethics and formal procedures of research and follows the typical qualitative research. It is used as a basis for policies related to the disabled in Germany and seems to have high research value to provide a foothold for clinical reasoning for clinicians.
However, at the same time, there are some weak points. Please read presented below and, if possible, I recommend reflect them in the paper to benefit readers who read this paper.
1. Ethical issue
The certification of research ethics has already been written in the paper, but at least qualitative research or phenomenological research should mention specifically how the core ethical issues of the photo-voice paper were dealt with. For example, it should be specifically described how the copyright and license issues that arise when the photographs are used in future research or policy meetings, or how the scope of the photograph's use was determined when the participants voluntarily agreed. Photography might cause privacy infringement of the participant himself or the subject, but how the details were assumed and consulted (not simply agreed)
2. Issue of the Research Method itself
This research method was introduced in the 1990s, and it is judged that somewhat behind the current era in which motion pictures-based media communication methods are mainstream. Others pointed out that a majority of the literature is filmed with pre-determined research topics, rarely consult with local communities, questions presented by researchers are too difficult to apply, so they are often given up, forgotten, or replaced with community participation before and after research. It is necessary to describe the responses to these parts or the authors' thoughts.
3. the cliches of the research subject
I strongly recommended authors to emphasize why this research is different compare with previous similar studies because this could be looked cliché for sometimes. The main theme of this study has already been studied in various ways for a long time, and can be regarded as a follow-up to existing studies.
4. academic formation
Of course, the diversity of research is very important to post in this journal, but the basic form of academic research must be established. The presented table does not need to be a quantitative figure, but it is not desirable to mark the table in a hybrid form (e.g. Scheme 2014.4 Photo Shooting Phase). Efforts should be made to enhance academic value overall. It is regrettable that there is almost no decision process to develop the six derivation processes decided by the authors academically and logically. If these parts are not reinforced, they may be misunderstood as newspaper articles or contributions.
Thank you for the research and Good luck.
Author Response

(The authors gave the same response as above.)

Round 2
Reviewer 3 Report
The authors' revised manuscripts shows signs of a lot of concern and a lot of actual corrections. There are still cliches of research topics and decreased academic research everywhere, but overall, I think we could show better research results if we refine it a little bit more. Thank you.
Author Response
Dear reviewer,
Thank you very much for the renewed evaluation of our article and the advice for further improvement.
We revised the manuscript according to your recommendations. Please see the attachment.
Sincerely yours,
On behalf of all authors,
Tabea Böttger
